# Dual Roles of the AMP-Activated Protein Kinase Pathway in Angiogenesis

**DOI:** 10.3390/cells8070752

**Published:** 2019-07-19

**Authors:** Yuanjun Li, Ruipu Sun, Junrong Zou, Ying Ying, Zhijun Luo

**Affiliations:** 1Jiangxi Provincial Key Laboratory of Tumor Pathogens and Molecular Pathology, Department of Pathophysiology, School of Basic Medical Sciences, Nanchang University Jiangxi Medical College, Nanchang 330006, China; 2Queen Mary School, Nanchang University Jiangxi Medical College, Nanchang 30006, China

**Keywords:** angiogenesis, tumorigenesis, retinopathy, AMPK, mTOR, TGF-β, VEGF, HIF-1α

## Abstract

Angiogenesis plays important roles in development, stress response, wound healing, tumorigenesis and cancer progression, diabetic retinopathy, and age-related macular degeneration. It is a complex event engaging many signaling pathways including vascular endothelial growth factor (VEGF), Notch, transforming growth factor-beta/bone morphogenetic proteins (TGF-β/BMPs), and other cytokines and growth factors. Almost all of them eventually funnel to two crucial molecules, VEGF and hypoxia-inducing factor-1 alpha (HIF-1α) whose expressions could change under both physiological and pathological conditions. Hypoxic conditions stabilize HIF-1α, while it is upregulated by many oncogenic factors under normaxia. HIF-1α is a critical transcription activator for VEGF. Recent studies have shown that intracellular metabolic state participates in regulation of sprouting angiogenesis, which may involve AMP-activated protein kinase (AMPK). Indeed, AMPK has been shown to exert both positive and negative effects on angiogenesis. On the one hand, activation of AMPK mediates stress responses to facilitate autophagy which stabilizes HIF-1α, leading to increased expression of VEGF. On the other hand, AMPK could attenuate angiogenesis induced by tumor-promoting and pro-metastatic factors, such as the phosphoinositide 3-kinase /protein kinase B (Akt)/mammalian target of rapamycin (PI3K/Akt/mTOR), hepatic growth factor (HGF), and TGF-β/BMP signaling pathways. Thus, this review will summarize research progresses on these two opposite effects and discuss the mechanisms behind the discrepant findings.

## 1. Introduction

Blood vessels deliver nutrients and exchange substances among different organs and systems. Formation of blood vessels is an essential process for organ development, which includes vasculogenesis and angiogenesis [1]. The process by which blood vessels are developed from progenitor cells is called vasculogenesis while angiogenesis is built on a previous blood vessel network. Angiogenesis plays essential roles in both physiological and pathological scenarios. Under such circumstances as hypoxia, strong contraction of muscle and wound healing, larger amount of oxygen is demanded and angiogenesis is triggered to compensate for oxygen deficiency. During this period, disruption of integrity of existing blood vessel walls, and proliferation and migration of endothelial cells are the two first steps to occur under the stimulation of cytokines including vascular VEGF) and TGF-β/BMPs. These cytokines upregulate enzymes like matrix metalloproteases (MMPs) to cause degradation of the blood vessel basement, and promote proliferation and migration of endothelial cells [2]. Newly formed blood vessels are composed of inner lining of endothelial cells and encircling pericytes, fibroblasts and smooth muscle that provide anatomical and functional stability.

The process of pathological angiogenesis is similar to that in physiological conditions, but endothelial cells and pericytes are somewhat different; for example, in cancer and possibly diabetic retinopathy, pericytes are loosely attached to vasculature, which is leaky and hemorrhagic, albeit convenient to deliver nutrients. This is partly attributed to overproduction of VEGF. Angiogenesis concurs with the growth of a solid tumor in coping with the increased need for oxygen and nutrients. Therefore, anti-angiogenesis has long been regarded as a strategy for cancer therapy, which is widely used alone or in combination with radiotherapy or chemotherapy [3].

AMPK is a heterotrimeric kinase that phosphorylates substrates at serine/threonine residues and is conserved in the eukaryote kingdom. AMPK serves as a fuel-sensing enzyme that is activated under metabolic stresses in which AMP or the ratio of AMP to ATP is increased, and suppressed in energy surplus [4]. AMP directly binds to the regulatory subunit of AMPK, which induces a conformational change of the holoenzyme, leading to its activation. Binding of AMP also enables phosphorylation of T172 by upstream kinases such as liver kinase B (LKB1) and calmodulin-dependent protein kinase kinase beta (CamKKβ) for maximal activation, and prevents inactivation by dephosphorylation [5]. AMPK can be activated by incubation of cells with agents that mimic AMP or cause the increase of intracellular AMP, such as 5-aminoimidazole-4-carboxamide ribonucleotide (AICAR) and metformin. Under stress conditions, activation of AMPK attenuates anabolic processes that consume ATP, preserving ATP for an acute cell survival program, and stimulates catabolic processes to replenish ATP [5]. Since many AMPK pharmacological activators such as metformin, berberine, resveratrol, and thiazolidinediones (TZDs), have been shown to lower levels of blood glucose and lipids, and used in clinic for the treatment of type 2 diabetes and obesity, AMPK is a well-received therapeutic target for metabolic diseases [6,7]. The role of AMPK in tumorigenesis and cancer progression has emerged as an important area in which extensive reviews can be found elsewhere [8,9]. With regard to the effects of AMPK and its activators on angiogenesis, it is less well defined and no conclusive findings are obtained. This review focuses on the aspects of AMPK that play both positive and negative roles in angiogenesis and discuss the potential connection of AMPK with some important regulators in angiogenesis.

## 2. Orchestrated Action of VEGF, Notch and TGF-β/BMP Signaling on Angiogenesis and Vascular Patterning

Increasing demand for blood supply during physiological and pathological processes such as embryogenesis, ischemia, exercise, wound healing, and tumorigenesis requires formation of new blood vessels, an event that is initiated by sprouting angiogenesis that stems from existing endothelial cells and is characterized by leading tip cells and trailing stalk cells [10]. Sprouting angiogenesis is a complex process that is regulated by concerted signaling pathways. It starts with a tip cell that moves along a gradient of VEGF, immediately followed by the growth of a trunk of endothelial stalk cells (Figure 1). A hypoxic microenvironment resulting from lack of blood supply creates gradients of VEGF-A, which then triggers migration of single endothelial cells, a process to select the leading tip cells that guide emerging sprouts [11,12]. The tip cells are essential at the start of sprouting by degrading the basement membrane and migrating through extracellular matrix. The elongation of the new vessels relies on the proliferation of stalk cells. Eventually, when the tip cells meet the tips of other sprouts, branches anastomose and lumenized vessels are then formed. The endothelial cells in the newly formed vessel are non-migratory and non-proliferative, and thus become quiescent phalanx cells. They are then organized in a continuous monolayer forming a tight barrier and tube that maintains blood flow, and partitions and exchanges materials between blood vessels and tissues.

Notch plays pivotal roles in selection of the trailing stalk cells behind the tip cells [10,13]. As illustrated in Figure 1, VEGF in the hypoxic environment stimulates expression of delta-like ligand 4 (DLL4) in the leading tip cells that binds to and activates Notch receptor in the adjacent trailing stalk cells, causing its cleavage, release from the plasma membranes and translocation into the nucleus (Notch intracellular domain, NICD). NICD triggers transcription of target genes including Hey1, Hey2, Hes1, leading to downregulation of VEGFR2, VEGFR3 and DLL4, and upregulation of VEGFR1. The downregulation of VEGFR2/3 maintains the phenotype of the stalk cells, while the downregulation of DLL4 minimizes the activation of Notch signaling in the tip cells [14,15,16]. VEGFR1 is a decoy receptor that binds VEGF with high affinity, but has low tyrosine kinase activity, and can also be converted to a soluble form. As a result, VEGFR1 competes with VEGFR2 for VEGF to inhibit angiogenic sprouting by limiting conversion of stalk cells to tip cells [17,18]. Thus, the resultant decrease of responsiveness to VEGF ensures the neighboring endothelial cells become proliferating stalk cells [13,19].

TGF-β/BMP induce phosphorylation and activation of Smad1/5/8, which then upregulate the inhibitor of DNA-binding proteins (Id1-3) and integrate signals with Notch in the regulation of stalk cell phenotype [20]. Id1 stabilizes Hes 1 by forming a complex, but is destructed by interaction with Hey1 [21]. As Hes1 suppresses the transcription of DLL4 and VEGFR2, fluctuation of Id levels in angiogenic sprouting plays an important role in shuffling between tip and stalk cells. Thus, high levels of Ids in the stalk cells maintain features of stalk cells, while reduction of Ids in a cellular context favors tip cell selection because of induction of Dll4 upon VEGFR2 signaling [20]. Furthermore, phosphorylated Smad1/5/8 directly regulates or cooperates with activated Notch to regulate transcription of the Notch target genes [21,22]. Therefore, the dynamics of tip-stalk cell shuffling in the sprouting angiogenesis is regulated by a concerted action of many signals, which undergoes oscillatory changes in physiological settings. Disruption of these signaling pathways will cause imbalance of the ratio of tip cells to stalk cells, leading to aberrant angiogenesis. For example, ablation of Smad1/5 in endothelial cells impairs Notch signaling, resulting in increased expression of tip cell markers such as VEGFR2/3 and DLL4 and decreased levels of stalk cell-enriched markers (e.g., Id1-3, Hes1, Hey1, Jagged1, and VEGFR1) [20]. Therefore, high VEGFR2-Dll4 levels in stalk cells as a result of impaired activin receptor-like kinase 1 (ALK1)-Notch coupled signaling drive the cells to acquire a tip-cell-like cell phenotype and to accumulate activated Notch receptors in the nuclei of the leading tip-cell-like cells. Hence, the complex phenotype in the absence of Smad1 and Smad5 in endothelial cells appears to be excessive sprouting and defective in directed cell migration due to the failure to balance the tip versus stalk cell ratio [20]. Another example is hereditary Hemorrhagic Telangiectasia (HHT), a class of genetic disease resulting from heterozygous mutations of *ALK1* or endoglin (ENG), a non-kinase accessory protein for ALK1 [23,24]. HHT is a familial human vascular syndrome that is characterized by development of fragile and direct connection between arteries and veins, or arteriovenous malformations. Recent studies have revealed excessive angiogenesis and endothelial “tip cells” at the vascular front that display migratory defects in retinas of *ALK1*^+/−^ mice [25]. VEGFR1 levels are reduced in *ALK1*^+/−^ mice and HHT2 patients, suggesting similar mechanisms in humans and mice with mutations of the *ALK1* gene.

The number of regulators that actually participate in the regulation of angiogenesis are far more than that listed above. Nevertheless, they are mainly involved in the blood vessel sprouting and patterning. Altered expression and activation of these regulators in association with diseases can lead to dysregulated angiogenesis or malformed blood vessels. Many oncogenes in solid tumor can be proangiogenic or exert indirect effect via acting on stroma cells, which then secrete proangiogenic factors that cause imbalance of the ratio of tip cells to stalk cells, resulting in aberrant angiogenesis. As such, the blood vessels in tumor are particularly perme-able, tortuous and greatly different from normal vasculature. Therefore, targeting altered regulators to normalize vasculature may attain the goal of cancer therapy [26].

## 3. The Role of Metabolism in Sprouting Angiogenesis

The role of glucose and lipid metabolism in sprouting angiogenesis has only recently been paid attention [27,28]. One would expect that endothelial cells adapt to an oxidative phosphorylation via metabolizing glucose in mitochondria, as these cells are exposed to high concentration of oxygen in the blood. To be surprised, however, 85% of their ATP is generated via glycolysis [29]. Although the efficiency of ATP production per one molecule of glucose is very low through glycolysis as compared to oxidative phosphorylation, it offers several advantages. For instance, glycolysis in endothelial cells enables them to vascularize avascular anoxic tissues, an advantage that oxidative phosphorylation is unable to offer. Secondly, glycolysis generates ATP quickly to compensate its low efficiency and provide necessary energy for sprouting, thereby rapidly restoring oxygen supply to the surrounding tissue [30]. Third, intermediate metabolites of glycolysis provide precursors of macromolecules needed for cell division and generate reducing power for redox homeostasis required for sprouting.

It has been shown that VEGF increases the expression of 6-phosphofructo-2-kinase/fructose-2, 6-bisphosphatase-3 (PFKFB3), a critical activator for glycolysis [29]. Inactivation of the *PFKFB3* gene in endothelial cells reduces glycolysis and impairs vascular sprouting by decreasing both migration of the tip cells and proliferation of the stalk cells, while overexpression of PFKFB3 stimulates glycolysis and promotes the tip cell phenotype during vessel sprouting [29]. Interestingly, modulation of PFKFB3 levels does not apparently alter the expression of tip or stalk cell markers, suggesting a direct effect of glycolysis [29].

In addition to the role of glycolytic flux, recent studies have demonstrated that fatty acid oxidation is also important for angiogenesis. Thus, inhibition of carnitine palmitoyltransferase 1a (CPT1a), a key enzyme that transports free fatty acids into mitochondria for beta-oxidation, perturbs proliferation of endothelial cells without affecting their migration or motility [31]. Furthermore, endothelium-specific deletion of CPT1a in mice results in impairment of postnatal vascular development in the retina manifesting decreases in vascular branch points and radial expansion of the vascular plexus. These alterations are attributed to reduced proliferation of the stalk cells with normal phenotype of the tip cells. The abnormality induced by disturbance of CPT1a is not caused by changes in ATP production, but by inhibition of DNA synthesis, in which the incorporation of fatty acid-derived carbons into aspartate and uridine monophosphate, both of which are precursors for de novo dNTP synthesis, is suppressed [31].

It has been reported that knockout or pharmacologically inhibition of fatty acid synthase (FSN) in endothelial cells impedes vessel sprouting by reducing cell proliferation, and angiogenesis in vivo [32]. Ablation of FSN elevates levels of malonyl CoA, which causes malonylation of mTOR at lysine 1218 and inactivation of mammalian target of rapamycin complex 1 (mTORC1).

AMPK has long been known to play essential roles in regulation of glycolysis and fatty acid oxidation [5]. The abovementioned metabolisms are also regulated by AMPK. First, AMPK can enhance glycolysis by phosphorylating and upregulating PFKFB3 [33]. It is possible that the upregulation of PFKFB3 by VEGF is mediated by activation of AMPK. In keeping with this notion, a previous report has shown that VEGF activates AMPKα1 through CamKKβ-dependent mechanism, which plays an essential role in angiogenesis both in vitro and in vivo [34]. Secondly, AMPK phosphorylates acetyl CoA carboxylase to diminish intracellular levels of malonyl CoA, leading to increased transporting of fatty acid into mitochondria for beta-oxidation [6]. As noted above, these two functions of AMPK may play a positive effect on sprouting by favoring tip cells and stalk cells, respectively. However, AMPK inhibits mTORC1 activity via multiple mechanisms [7], which may exert a negative effect on angiogenesis, similarly to that caused by the inhibition of FASN. How and when positive and negative regulation of angiogenesis by AMPK occur warrants clarification.

## 4. Positive Regulation of Angiogenesis by Adenosine 5′-Monophosphate-Activated Protein Kinase (AMPK)

The study on the positive role of AMPK originated from its protective effect on endothelial cells in the metabolic syndrome, ischemia and hypoxia (Figure 2). Atherosclerosis, and myocardial infarction are commonly associated with type 2 diabetes [35]. These conditions can cause damages to endothelial cells and thickness of blood vessel walls, thereby leading to local hypoxia in the pathologic areas. Thus, AMPK plays positive roles in several aspects. First, it exerts protective effects on endothelial cells against high glucose and fatty acid insults [36,37,38,39,40,41,42]. Second, AMPK stimulates differentiation of endothelial progenitor cells, proliferation and migration of endothelial cells [43,44]. Third, AMPK acts as an upstream kinase of endothelial nitric oxide synthase such that nitric oxide (NO) production is increased, resulting in vasodilation of blood vessels and angiogenesis [45,46]. Fourth, AMPK activation under hypoxia facilitates autophagy, which somehow stimulates expression of VEGF [47,48,49].

The early study showing the positive role of AMPK in angiogenesis is that hypoxia induces the activation of AMPK in human umbilical vein endothelial cells (HUVEC) [49]. Suppression of AMPK inhibits migration of HUVEC to VEGF and tube formation in vitro under hypoxic but not normoxic cultures and blunts angiogenesis in matrigel plug in mice [49]. Similar findings were reported using adiponectin as a proangiogenic factor [50]. Interestingly, these studies placed AMPK upstream of the PI3K/Akt signaling pathway. Furthermore, studies from the same group have demonstrated that LKB1/AMPK regulates angiogenesis in response to ischemic injury in hind limbs of mice [51,52]. Many of later studies have corroborated the positive role of AMPK in angiogenesis or protective effects on endothelial cells under such circumstances as hypoxia, ischemia, stroke, and oxidative stress, and in response to cytokines and pharmacological agents such as AICAR, metformin and statins [34,43,44,45,53,54,55,56,57,58,59,60,61,62,63,64,65,66,67,68,69].

## 5. Negative Effects of AMPK on Angiogenesis

A considerable number of studies have also documented the negative effect of AMPK on angiogenesis (Figure 3). One incidence is its protective effects in diabetic retinopathy, one of the most serious diabetic complications and the first leading cause of blindness, and age-related macular degeneration as well. The initial pathological changes in diabetic retinopathy include retinal neuropathy and retinal vasculopathy. High glucose and lipids exerts glucotoxicity and lipotoxicity to retinal neurons, leading to their apoptosis and thinning of the retinal neuronal layers [70]. These are followed by obstruction of retinal arterioles and angiogenesis. Abnormal angiogenesis causes irregular retinal vasculature resulting in edema, hemorrhage, and finally retinal detachment. AMPK has been shown to play protective roles in retinopathy. First, AMPK protects retinal neuron, glial cells and retinal pigment epithelial cells induced by oxidative stress and inflammation [71,72,73,74,75,76]; second, AMPK activation exerts a dilating effect on retinal arterioles to improve circulation in narrow arterioles [77,78]; third, AMPK inhibits angiogenesis, so as to minimize edema and hemorrhage [79,80,81,82,83,84]. Studies have shown that AMPK activation promotes nuclear enrichment of AMPK α2 subunit and suppresses HIF-1α and VEGF [79] or VEGF receptor Flk-1, leading to inhibition of angiogenesis in oxygen-induced retinopathy [80]. A similar conclusion has been drawn using diabetic mouse model that TDZs reduce oxygen-induced neovascularization in ischemic retinal through the adiponectin-dependent mechanism, suggesting a role of AMPK in the protection [81]. In addition, it has been documented that AMPK activation leads to inhibition of choroid neovascularization, a pathogenic factor in age-related macular degeneration [82,83].

The precise role of AMPK in cancer development and progression is controversial. Many studies have shown that AMPK activation exerts an inhibitory effect on cancer cell growth and progression, while others have claimed its promoting effect [7,8]. Metformin is the first line of medicine in the treatment of type 2 diabetes and has been shown to be beneficial to cancer patients and inhibit cancer growth in vitro and preclinical studies [85,86,87,88]. As angiogenesis is prerequisite for cancer development, investigations have been carried out to look into the effect of AMPK. In fact, many studies have described that activation of AMPK by a variety of pharmacological activators plays an inhibitory role in tumor angiogenesis. For example, activation of AMPK by metformin, AICAR and sivastatin attenuates tumor growth along with angiogenesis [84,89,90,91]. Secondly, hop-derived flavonoid xanthohumol exhibits potent angiopreventive activity, which is mediated by the activation of AMPK and inhibition of eNOS phosphorylation, thereby reducing production of NO in endothelial cells [92]. Thirdly, itraconazole, a clinically used antifungal drug, possesses potent antiangiogenic and anticancer activity [93]. It was found to inhibit the mitochondrial protein voltage-dependent anion channel 1 (VDAC1) that controls passage of ions and small molecules through the outer membrane of mitochondria and thus regulates mitochondrial metabolism. Inhibition of VDAC1 by itaconazole leads to increases in AMP to ATP ratio, followed by activation of AMPK. Fourthly, N6-isopentenyladenosine (iPA), an end product of the mevalonate pathway with an isopentenyl chain, exerts a tumor suppressive effect against various tumors. iPA is converted to 5′ monophosphorylated form (iPAMP) by adenosine kinase inside cells, which inhibits angiogenesis via direct activation of AMPK [94].

## 6. The Mechanisms Underlying the Opposite Effects of AMPK on Angiogenesis

There may be several reasons for the discrepant results regarding the effect of AMPK on angiogenesis. Whether AMPK promotes or inhibits angiogenesis depends on physiological or pathological settings (Figure 2 and Figure 3). Under stresses such as ischemia, hypoxia, and energy deprivation, AMPK is activated by increased level of AMP to ATP ratio, or through AMP-independent mechanism engaging lysosomes [95]. AMPK activation has been known to have a positive effect on autophagy by inhibition of mTOR and direct phosphorylation of autophagy modulators [96]. Autophagy somehow leads to stabilization of HIF-1α [97]. HIF-1α then regulates VEGF and possibly other factors that altogether facilitate formation of new blood vessels to improve local circulation and meet the need for nutrients. In this circumstance, the newly formed blood vessels contain mature and regular structure. In contrast, on pathological conditions, for example, in diabetic retinopathy, high glucose can suppress AMPK activity and activate mTOR. The latter upregulates HIF-1α, leading to the increased expression of VEGF [7,98,99,100]. Under this circumstance, irregular blood vessel structure is formed, resulting in edema and hemorrhage. Thus, activation of AMPK and inhibition of mTOR should attenuate pathological angiogenesis.

Tumor angiogenesis is far more complicated, as it is regulated by many factors. First, at early stages when a cancer cell cluster expands its mass before building up blood vessels, a hypoxic environment can stimulate the expression of VEGF, which leads to extension of blood vessels from adjacent normal tissues to replenish blood and nutrient supplies. The way in which AMPK is activated and exerts a protective effect at this step via promoting autophagy has been described. Conceivably, AMPK might also stimulate angiogenesis when autophagy exerts a protective effect [101]. Second, many oncogenes are proangiogenic. For example, the PI3K signaling pathway is frequently activated due to gene mutations of PI3K, loss-of-functional mutations of phosphatase and tensin homologue (PTEN), or overexpression of growth factors, cytokines, or gene amplification of its upstream regulators such as human epidermal receptor 2 (HER2). The activated PI3K/Akt can lead to activation of mTOR, which upregulates HIF-1α and VEGF. Correlation between the levels of tyrosine-phosphorylated HER2 and HIF-1α/VEGF has been reported [102]. Conditional culturing of endothelial cells with HER2 positive breast cancer cells enhances tube formation in vitro, which is diminished by the treatment with metformin [102]. The inhibitory effect of metformin on angiogenesis was observed in allograft tumor model using mouse 4T1 cancer cells. In an animal study of 1,2-dimethylhydrazine-induced colon cancer, it was found that the expression of IGF-1 is increased, and metformin downregulates IGF-1 and suppresses tumor angiogenesis [103]. Therefore, it appears that many of these signals funnel to activation of mTOR-HIF-1α-VEGF axis. As cancer cells are addicted to this dysregulated axis for their growth, metformin or other AMPK activators as discussed above can disrupt them, leading to the inhibition of angiogenesis.

The process of angiogenesis is regulated by changes in the balance of pro-angiogenesis and anti-angiogenesis. VEGF is the most studied proangiogenic cytokine that is frequently overexpressed in cancers and can be regulated by many signaling pathways [104]. Among them is the TGF-β/BMP that can upregulate VEGF, leading to sustained angiogenesis by stimulating proliferation, differentiation and migration of endothelial cells [105,106,107]. Thus, the TGF-β/BMP signaling pathways have been regarded as therapeutic targets for cancer progression and angiogenesis [108,109]. The TGF-β and the associated signaling pathway exert a tumor suppressive function at early stages. Once cancer cells escape its inhibitory effect at later stages, expression of TGF-β is increased and exhibits pro-metastatic and proangiogenic effects [108]. Studies have demonstrated that TGF-β in low concentrations interacts with the ALK1 receptor and results in increased expression of the metalloproteases (e.g., MMP-2 and MMP-9), and enhances the migration of endothelial cells, while in higher concentrations, TGF-β interacts with ALK5 receptor and hinder angiogenesis [110].

Most BMPs are proangiogenic except that BMP-9 and BMP-10 are anti-angiogenic. BMP-6 is highly expressed in higher grade, primary and advanced prostate cancer with metastasis [111,112]. Hepatocyte growth factor (HGF) and its receptor, c-Met, have been shown to play an important role in cancer angiogenesis [113,114]. HGF upregulates BMP-7 and BMP receptors, BMPR-IB and BMPR-II, in prostate cancer cells, and the BMP co-receptor (RGMb) in vascular endothelial cells [115,116,117]. These studies suggest that HGF and BMP-7 may work together to enhance angiogenesis. In addition, BMP can synergize with VEGF, TGF-β, bFGF, and PDGF-BB to promote angiogenesis [118,119,120].

Previous studies have demonstrated that AMPK inhibits the TGF-β signaling pathway [121,122,123,124,125]. Furthermore, studies have illustrated an inhibitory role of AMPK in cancer cell migration and EMT via both TGF-β-dependent and -independent mechanisms [122,123,124,126,127,128,129,130]. Recently, we have shown that metformin inhibits Smad2/3 phosphorylation and expression of target genes involved in EMT, metastasis, and angiogenesis, including PAI-1, CTGF, as well as IL6 [131]. These effects are mainly mediated by activation of AMPK [131]. AMPK also suppresses the production of TGF-β1 in cancer cells [132]. Consistently, the serum levels of TGF-β1 in patients with type 2 diabetes receiving metformin are lower than those taking other glucose lowering medicine. Interestingly, the inhibitory effect of AMPK on TGF-β signaling is not just limited to Alk5/Smad2/3 paradigm, but also extends to BMP/ALK1/2/Smad1/5 signaling [82,133]. We have recently found that metformin inhibits ALK1-mediated tube formation and angiogenesis in matrigel plug assay [133]. The inhibitory effect is possibly mediated by proteosomal degradation of the ALK receptor involving Smurf-1/Smad6 interaction [133]. Furthermore, our study has shown that choroid neovascularization induced by laser photocoagulation is inhibited by metformin, which correlates with the reduction of ALK1 [82]. Altogether, these studies highlight targeting ALK1 can inhibit angiogenesis and tumorigenesis complementary to anti-VEGF therapies and activation of AMPK can have a double hit effect [134].

## 7. Conclusions

AMPK acts as a fuel gauge that controls energy homeostasis at cellular levels, while angiogenesis provides assurance to maintain systemic energy balance in human body. In an energy crisis resulting from hypoxia, ischemia, stroke and exercise, AMPK is activated and exhibits protective effects, including promoting angiogenesis. By contrast, under pathological conditions, such as retinopathy and cancer, AMPK is inhibited. Oncogenic factors that are usually proangiogenic are hyperactive in cancer, leading to aberrant angiogenesis. Many of these proangiogenic factors can be inactivated by sustained activation of AMPK. To be noteworthy, most of the conclusions on the effects of AMPK on angiogenesis are obtained from in vitro and animal studies. As many AMPK activators such as metformin, aspirin, berberine, and resveratrol have already been used in clinics for diseases without overt side effects, it is rational and feasible to use them in clinical trials in treating issues with angiogenesis under both physiological and pathological conditions. We hope to see their beneficial effects through clinical trials in the near future.

## Figures and Tables

**Figure 1 cells-08-00752-f001:**
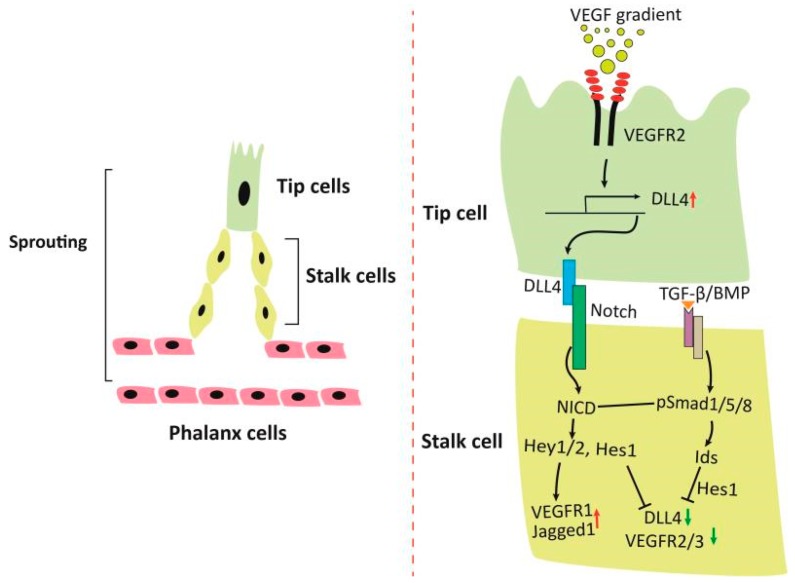
Regulation of sprouting angiogenesis. A gradient of VEGF leads to migration of tip cells and upregulates Dll4, which then activates Notch in the adjacent stalk cells. Expression of Notch target genes maintains phenotype of the stalk cells. TGF-β/BMP signaling cooperates with Notch signaling and regulates sprouting angiogenesis. Abbreviations: Dll4, Delta Like Canonical Notch Ligand 4; Jagged1, a Notch ligand encoded by *JAG1*; NICD, Notch intracellular domain; TGF-β/BMP, transforming growth factor-beta/bone morphogenetic protein; Smad1/5/8, transcription factors downstream of TGF-β/BMP; Hey1/2, hairy and enhancer of split-related with YRPW motif protein 1 and 2, Notch target proteins; Hes1, hairy and enhancer of split-1, a Notch target protein; Ids, inhibitor of DNA binding proteins; VEGF, vascular endothelial growth factor; VEGFR2/3, VEGF receptor 2 and 3.

**Figure 2 cells-08-00752-f002:**
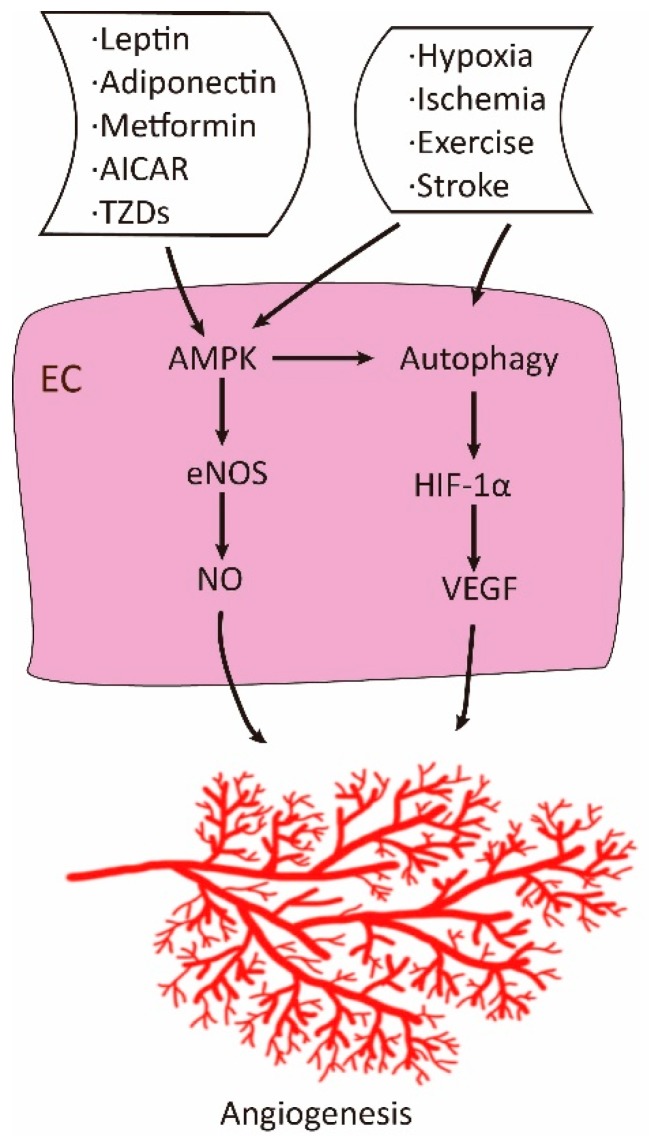
Positive role of adenosine 5′-monophosphate-activated protein kinase (AMPK) in angiogenesis. Under physiological conditions, AMPK is activated by ligands, pharmacological agents, and stresses. Activated AMPK promotes angiogenesis via increased production of NO and vascular endothelial growth factor (VEGF).

**Figure 3 cells-08-00752-f003:**
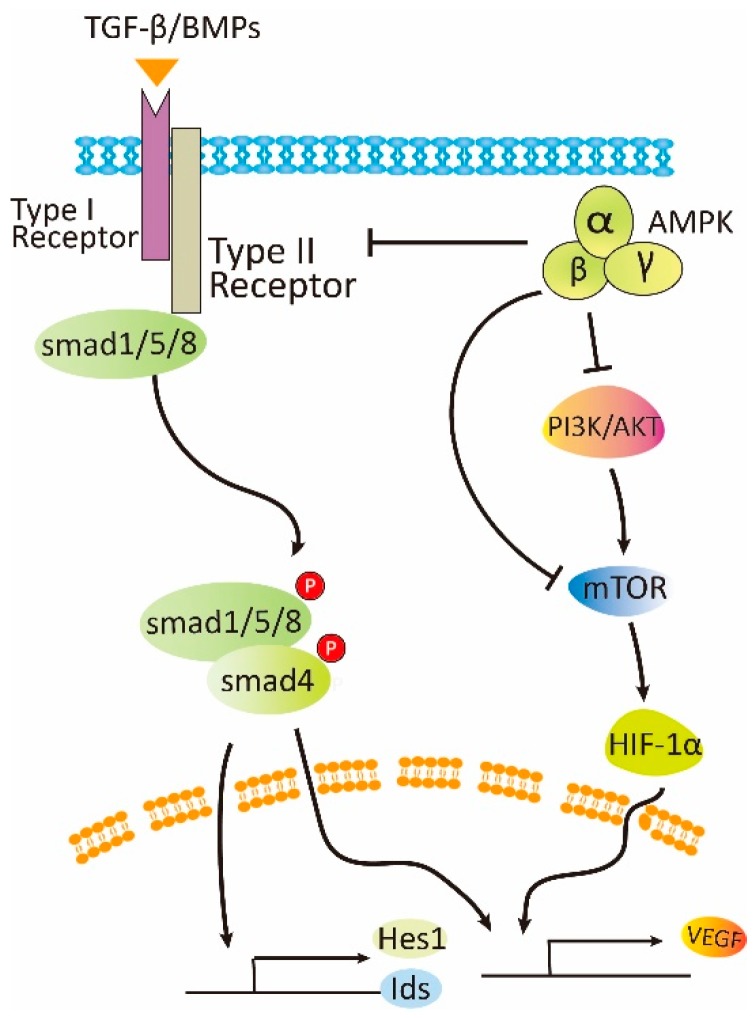
Negative role of AMPK in angiogenesis. Under pathological conditions, AMPK can attenuate angiogenesis by inhibition of mTOR and TGF-β/BMP signaling.

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
