# Peer review of "Dual Roles of the AMP-Activated Protein Kinase Pathway in Angiogenesis"

_cells, 2019, doi:10.3390/cells8070752_

Reviewer 1 Report

In this mini-review, authors described the dual role of AMPK signaling in angiogenesis. They give a compressive overview of the mechanisms of angiogenesis by illustrating some of the previous reviews on VEGF, Notch signaling and TGFbeta in Tip and Stalk cells. The role of AMPK in normal and pathological conditions is well illustrated and its link with HIF and mTOR signaling is presented and controversies on the pro and anti-angiogenic effects of AMPK are discussed.

For improvement, authors should clarify the figure 1, as the draw is quiet confusing when talking about Tip and Stalk cell interactions. While this is well described in the main text, I would suggest to draw a vessel with Tip and Stalk cells et clarify how VEGF gradient and signaling in Tip cells regulate Notch in Stalk cells and how it affect their migration and differentiation during angiogenesis.

The last sentence (lines 117-120) describing the deletion of smad1 and Samd5 in endothelial cells should be supported by a reference and clarify in which model or pathology the lack of TGF beta signaling through Smad1/Smad5 or Smad2/3 resulted in a chaotic vessels. This is well described in Ehlers-Danlos syndrome.

Before the positive regulation of angiogenesis by AMPK (line 130), it will be useful to describe in a new paragraph the role of metabolism in regulating angiogenesis. There is a large body of evidence from Carmeliet lab showing the role of PFKFB3, FAO and control of dNTP in vessel sprouting and branching. This will help to make a connection with AMPK.

Minors:

Line 33: add space between vasculogenesis and while

Line 40: add line between matrix and metalloproteases

Line 69: remove “the” between potential and connection

Line 250: replace cancer angiogenesis to “tumor angiogenesis”

Line 260: AMPK also suppresses the production of “not and” TGFbeta1 in cancer cells

Author Response

We thank this reviewer for constructive suggestions.  We would like to respond to critiques as follows:

1.     For  improvement, authors should clarify the figure 1, as the draw is quiet  confusing when talking about Tip and Stalk cell interactions. While this  is well described in the main text, I would suggest to draw a vessel  with Tip and Stalk cells et clarify how VEGF gradient and signaling in  Tip cells regulate Notch in Stalk cells and how it affect their  migration and differentiation during angiogenesis.

Response:  We made new figures.  As the scheme aims to summarize text and help  readers to understand the text, we think it should not be too  complicated.  We hope it is now improved.

2.     The  last sentence (lines 117-120) describing the deletion of smad1 and  Samd5 in endothelial cells should be supported by a reference and  clarify in which model or pathology the lack of TGF beta signaling  through Smad1/Smad5 or Smad2/3 resulted in a chaotic vessels. This is  well described in Ehlers-Danlos syndrome.

Response: We added the reference.  We also added brief description of a genetic disease, hereditary Hemorrhagic Telangiectasia, which is more relevant to sprouting angiogenesis, lines 153-162. 

3.     Before  the positive regulation of angiogenesis by AMPK (line 130), it will be  useful to describe in a new paragraph the role of metabolism in  regulating angiogenesis. There is a large body of evidence from  Carmeliet lab showing the role of PFKFB3, FAO and control of dNTP in  vessel sprouting and branching. This will help to make a connection with  AMPK.

Response: We thank this wonderful suggestion.  We add the section “The role of metabolism in sprouting angiogenesis”.

Reviewer 2 Report

The manuscript entitled “Dual roles of the AMP-activated protein kinase 3 pathway in angiogenesis” aims to summarize the research progresses on the dual roles of the AMP-activated protein kinase and discuss the mechanisms behind the discrepant findings. My specific comments are described below: 1. The figures and schemes published were not informative and didactic. In a previous publication of this research group (Gao et al., 2018, 50(6), 523–531, Acta Biochim Biophys Sin), there was a better representation of figures and schemes. Thus, I strongly suggest authors to provided new figures and schemes using this previous publication layout. 2. The AMPK pathway proteins have strong influence on angiogenesis and these proteins can be target for antiangiogenic therapies. Thus, authors could provide a topic with specific information regarding the therapeutic potential of AMPK proteins. 3. I realize that authors bring information regarding some drugs that can modulate AMPK pathway (i.e. metformin). However, in conclusion section, authors highlight the therapeutic importance of this pathway. Conclusion section “Hence, clinically used pharmacological activators of AMPK can be repurposed for treating issues with angiogenesis under both physiological and pathological conditions.” Thus, in this reviewer opinion, will be useful providing a specific topic in this review regarding drugs that can be used based on blocking or activation of AMPK pathway and a discussion of its application in future clinical trials. How far we are to use this agents in clinical practice? 4. Abstract page 1, lines 11-13. Please consider using “...tumorigenesis and cancer progression, diabetic retinopathy and age-related macular degeneration.” 5. There are some typos through the manuscript that need to be addressed. i.e. Page 2, lines 73-74: “... healing , and”. Please, consider revising the manuscript carefully.

Author Response

We thank this reviewer for constructive suggestions.  We would like to respond to critiques as follows:

1.     1. The figures and schemes published were not informative and didactic. In a previous publication of this research group (Gao et al., 2018, 50(6), 523–531, Acta Biochim Biophys Sin), there was a better representation of figures and schemes. Thus, I strongly suggest authors to provided new figures and schemes using this previous publication layout.

Response:  In this revision, we made new figures and tried our best to make them concise and clear.

2.     The AMPK pathway proteins have strong influence on angiogenesis and these proteins can be target for antiangiogenic therapies. Thus, authors could provide a topic with specific information regarding the therapeutic potential of AMPK proteins.

3.     I realize that authors bring information regarding some drugs that can modulate AMPK pathway (i.e. metformin). However, in conclusion section, authors highlight the therapeutic importance of this pathway. Conclusion section “Hence, clinically used pharmacological activators of AMPK can be repurposed for treating issues with angiogenesis under both physiological and pathological conditions.” Thus, in this reviewer opinion, will be useful providing a specific topic in this review regarding drugs that can be used based on blocking or activation of AMPK pathway and a discussion of its application in future clinical trials. How far we are to use this agents in clinical practice?

Response: Although there was strong evidence showing the regulatory roles of AMPK in angiogenesis, all of the data are obtained from in vitro and animal studies, none was shown in human studies.  We hope this review can help to stimulate clinical trials in the near future.  We added this notion to discussion in Conclusion.

4.     Abstract page 1, lines 11-13. Please consider using “...tumorigenesis and cancer progression, diabetic retinopathy and age-related macular degeneration.”

Response: Well taken.